# Conflicting 'mother-scientist' roles. An innovative application of basket analysis in social research

Ewa Krause[1], Renata Tomaszewska[1], Aleksandra Pawlicka[2]*

1 Kazimierz Wielki University in Bydgoszcz, Bydgoszcz, Poland, 2 University of Warsaw, Warsaw, Poland

* oliko.aleksandra@gmail.com, a.pawlicka5@uw.edu.pl

**Data Availability Statement:** The anonymized data is available from the Harvard Dataverse repository: https://dataverse.harvard.edu/dataset.xhtml?persistentId=doi:10.7910/DVN/ZWJTUC.

## Abstract

The article discusses the issue of career and motherhood of female scientists, which is part of a broader thematic area known as Work-Life Balance. The theoretical part refers to the social role theory and the mutual influence of work and career on family and motherhood. The situation of women scientists is presented, for whom fulfilling the role of a mother is an important, although natural barrier on the road to a scientific career. Previous analyses present in the literature revealed that for the vast majority of mothers-scientists, motherhood is a factor that significantly delays their plans related to the development of a scientific career. The paper presents the results of empirical research conducted on the basis of classical academic methodology. Then, based on the data obtained from 334 mothers-scientists, an innovative, multidisciplinary experiment using data mining solutions was conducted, to answer the research question: Is the basket analysis tool able to find possible correlations between the factors characterising the respondents, and the types and dimensions of conflict occurring between scientific career and motherhood they experience? The paper shows that, according to the study results, most respondents declare they indeed experience the conflict between the roles of a mother and a scientist. The most frequently declared dimension of the conflict is the time-related one, then subsequently the emotional dimension, and lastly the financial dimension; many scientists declare they experience more than one dimension of conflict. Lastly, the basket analysis tool objectively confirmed the occurrence of correlations between the factors characterising the respondents and the types and dimensions of conflict occurring between scientific career and motherhood.

## Introduction

Both women and men take on and perform a variety of roles in society. Women generally function within two main social roles: family and work-related ones. The former includes the mother/maternal role and the latter is related to work and career—in the case of the female scientists, to a scientific career.

Family and professional roles of women are often considered in parallel, in order to highlight the difficulties and problems faced by professional women. In the modern world, work

**Funding:** The authors received no specific funding for this work.

**Competing interests:** The authors have declared that no competing interests exist.

has become a type of activity which colonises ever broader areas of life. On the one hand, it has become a space of emancipation and self-realization, especially for women; on the other hand —a field of intersection and conflict of various social roles. Work and career do have a considerable influence on family life and motherhood, as they are connected with lifestyle decisions, individual choices and professed values.

Female scientists are a professional group that seem to experience various forms of the conflict between their family and professional roles; however, not many studies of this issue have been conducted yet. Thus, the authors of this paper wished to scrutinize whether the female scientists employed at academic institutions in Poland in fact experience the conflict between scientific career and motherhood. They also examined what the dimensions of the conflict between academic career and motherhood are, according to the surveyed women employed at academic institutions in Poland. Lastly, in the course of an interdisciplinary experiment, they employed a data mining technique to analyze the obtained data again.

The main contribution of this paper, which presents answers to three research questions (the aim of the study was achieved), is that apart from the classical academic methodology, i.e., the following methods: critical analysis of the literature, diagnostic and statistical survey, it has used multi- and interdisciplinary methodology supported by data mining techniques. The combination of traditional academic research methodology with the use of data-mining tools created an innovative and original experimental setup. In order to make the study replicable, the specific setup and description of the dataset is given. The presented experiment with the use of the data-mining technique is innovative, as no attempts to apply the so-called basket analysis to the issues of women's careers and motherhood have been reported to date. This study was the first of its kind in Poland. Also, no data on similar studies from other countries in the world have been found.

This paper is structured as follows: firstly, the extensive background and the theoretical framework for the study is presented, based on a literature review. Then, the experimental setup and study design are given. Subsequently, the paper presents the results of the first part of the study, followed by the basket analysis of the obtained data. The paper closes with a discussion of the results and the final conclusions.

## Background and theoretical framework

The social situation of the women scientists that are discussed in this article seems to be specific. Scientific career is a type of professional career. Even a brief review of the most important literature on this issue indicates the disproportion in reaching its highest stages, compared to men. The reason for this state is mainly motherhood. According to the literature search:

- For the vast majority of mothers-scientists, motherhood is a factor that significantly modifies (delays) their plans related to the development of a scientific career [1];

- Motherhood is the most important factor causing women to withdraw from scientific careers or limiting their pace [2];

- Women scientists' decisions to procreate are often postponed for years, especially after obtaining a doctoral or postdoctoral degree, which often means embarking on late motherhood with the risk of adverse consequences [3];

- Women who are successful in science and have achieved a stable position in academia are more likely than men to be single and childless [4].

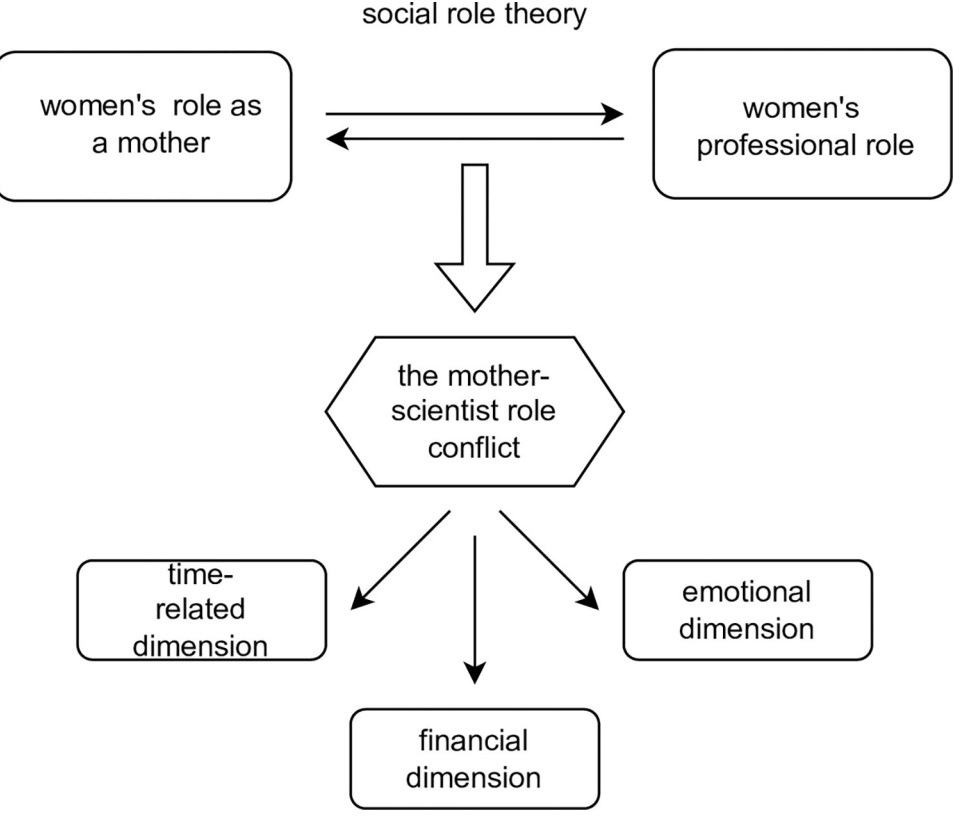

**Fig 1. Theoretical framework for the study.**

As shown later in the paper, although the number of women in Europe pursuing scientific careers is slowly increasing, they are still significantly underrepresented and their potential is not fully recognized and valued. This situation also applies to Poland.

Female scientists are an example of a professional group that pays a high personal price for the pursuit of professional ambitions—many of them are childless, and/or divorced, and single. Many of them choose not to become mothers for fear that they will not be able to cope with the pursuit for a PhD or a postdoctoral degree. Since in Poland every fourth mother-scientist indicates that family responsibilities are a hindrance to a scientific career (according to the study "Careers of young scientists" [5, 6]), this article adopts the thesis that motherhood may be a factor delaying the development of women's scientific careers, i.e. it may hinder them and be an obstacle to their professional work at universities or colleges. The aim of the empirical study was to determine the type and dimensions of conflict occurring between the field of scientific career and motherhood among female researchers employed in academic universities.

The theoretical framework behind the study has been presented in Fig 1:

## The conflict of the mother-scientist roles and its dimensions

It was assumed that the conflict of the mothers carrying out scientific, research and teaching activities can be described as "the conflict of the mother-scientist roles".

Such a formulation was already used in the study entitled "Functioning of mothers of scientists in the area of family and professional life" [7]. This conflict was defined as a situation occurring when the demands placed on a woman by her scientific work and by raising a child

in the role of a mother make it impossible for a given woman to fulfil the goals set for a scientist (meeting the formal conditions of work at a university) or, respectively, being a mother (e.g., providing a livelihood, creating a safe place for growth both physically and emotionally, creating and maintaining a relationship with the child developing towards respect and trust). The mother-scientist role conflict highlighted in the cited publication relates to the following three dimensions:

1. the time dimension, which includes lack of time to conduct scientific research; physical and psychological exhaustion related to child rearing; child-related responsibilities significantly limiting time for scientific work;

2. the financial dimension, which refers to the increased financial needs associated with an additional family member; abandonment of research in favour of teaching work; insufficient satisfaction of financial needs;

3. the emotional dimension, which refers to the stress of reconciling professional and domestic responsibilities; frustration at the overlapping of the roles of a mother and a scientist, the lack of time to build a relationship with the child, the pain of being separated from the child [7].

It is worth noting that the above-mentioned review of the literature on this issue indicates only a few comprehensive studies on the scientific careers of women and their motherhood as well as studies containing statistical data depicting the scale of the phenomenon of the conflict between the roles of mother and scientist (which the authors describe in greater detail further). So far, science has dealt only marginally with the impact of motherhood on the careers of women scientists, as well as with the effects of this impact from the point of view of both the scientists and the universities that employ them. Therefore, any attempt to include this issue in the scientific discourse is legitimate and necessary—especially in the form of scientific studies based not only on theoretical analyses, but primarily on the results of empirical research—and this is the case with this article. The presented research results are significant as they allow one to learn about the situation and experiences of female scientists in the context of combining motherhood and scientific career; to show their perspective. Importantly, the situation of women in academia in Poland is comparable to that of women in other European countries, which is discussed further in the article. Therefore, the results presented in this paper constitute a basis for further research analyses conducted by the authors in this problem area and may serve as a source of inspiration for other national and foreign researchers, e.g. in the field of identifying solutions limiting the conflict between the roles of mother and scientist that can be implemented at universities and support women in combining scientific development with fulfilling their parental role. Continuation of the research is needed especially due to the phenomenon of women disappearing/leaking out at successive stages of scientific careers ("leaky pipeline") and thus limiting the waste of their potential and talents. The presence of women in the sector of science and higher education, including in positions which enable making important decisions, is necessary to maintain the effectiveness of the education and science system, as well as to maintain high standards of scientific research. Therefore, it is necessary to diagnose the combination of a scientific career and motherhood not only on a national, but also on a European scale, and to introduce supporting solutions, requiring the needs of female scientists and mothers-researchers to be taken into account by the universities employing them.

The main areas among which adults divide their time and energy are work, family and parenthood. The number of people involved in the daily negotiation of responsibilities in many spheres of life is increasing. This applies to both women and men. The phenomenon of the accumulation of roles in the contemporary world coexists with such demographic, social and

cultural changes as: the increase in the rate of women's professional activity; the increase in the number of families in which both spouses are professionally active; the increase in the number of single parents; the ageing of societies and the intensification of the need to care for the elderly; changes in the social roles of women and men; as well as the increase in the standards of performing professional and family roles. These phenomena are associated with men's greater involvement in family life and childcare, women's greater involvement in professional work and employers' greater interest in their employees' quality of life [8].

## The social role theory

According to the social role theory, which has constituted the broadest theoretical framework for this work, an individual performs a whole set of roles, which is a system generating internal conflicts and requiring individual mechanisms of tension reduction. If an individual is not able to cope with the pressures and demands resulting from their set of roles, they may e.g.:

- assign unequal importance to the various pressures and demands of the role and consequently fulfil only some obligations;

- transfer their various responsibilities to others;

- limit or terminate the relationships and relations that are a source of the tension they feel [9–14].

The problem of the conflict between roles emerged more than 50 years ago [15]. Back then, it was observed that the role demands of participation in a particular organization conflict with the need to participate in other social groups [16, 17]. Based on this observation, inconsistent pressures within the family and work, mainly affecting women, were identified. It was found that women who are trying to meet the demands of both roles are subject to constant tension, as the need to perform one role is an obstacle to the effective performance of the other. Reconciling work and family responsibilities is a major challenge for many women and is often a source of conflict, which takes two forms: 1) work versus family (W-F), when the demands of the work role impede the fulfilment of the primary roles in the family, i.e., wife/partner and mother; 2) family versus work (F-W), when family responsibilities impede the fulfilment of the work role [18, 19].

Importantly, nowadays, the demands of family life and work life tend to be increasingly in conflict, and the interaction between the family process and work and career is by no means of one-way character. Family circumstances have an impact on what happens in the workplace, often resulting in impaired productivity. At the same time, family life and maternity issues are also affected by events in the professional field. The ability to carry out multiple roles can protect against the negative consequences of overload or conflict, as well as the cumulative side effects of these. Therefore, different life roles do not have to be mutually exclusive. On the contrary, instead of conflict, the phenomenon of facilitation may occur, which means that involvement in one role may facilitate involvement in another. This is because the multitude of assumed roles increases the possibility of gaining experience in different areas of life, and thus allows for a conflict-free transition from one form of activity to another. Such flexibility in the realisation of various social roles is a manifestation of the criterion of a person's personal maturity [8, 20, 21].

More specifically, in scientific literature, the issue of social roles, including career versus motherhood, is mainly referred to the Work-Family Fit; Work-Family Balance; or more broadly Work-Life Balance [22]. In the process of theoretical analysis of the interaction between work and career on family and motherhood, two main theoretical perspectives are adopted: negative impacts and positive impacts [21, 23–25].

The first perspective is described using terms such as conflict, strain, role conflict, overload, mismatch, negative radiation. This perspective is conceptualised on the basis of Stevan E. Hobfoll's gender role theory [26], and Robert Karasek's demands-control model [27]. The forerunners of the work-family conflict issue are Jeffrey H. Greenhaus and Nicholas J. Beutell [16, 28, 29], who defined it as inconsistent demands at work and at home, which in effect may lead to impeded performance of one role by involvement in the other role. These researchers focused on the impact of work and career on an individual's functioning in the family, analysing the conflict between these two areas in three dimensions:

- the amount of time devoted to them, which refers to the time pressure caused by physical involvement in one role at the expense of the other, as well as the pressure of psychological involvement when the individual, while fulfilling the other role, still is deep in thoughts about the matters related to the first one,

- the tension dimension, which refers to the situation in which the stress experienced while performing one of the roles makes it difficult to function effectively in the other role,

- the dimension of behaviour connected with a given role described as the lack of compatibility of behaviour styles in particular roles.

The second perspective—positive influences—is much less frequently undertaken in research. It emphasises the mutual reinforcement of roles, the possibility of improving the functioning in one area thanks to undertaking activity in another area. The terms used in this case are: positive radiation [30]; reinforcement [31]; enrichment [29]; and finally, facilitation [32], meaning the facilitation of meeting the requirements in a certain area of functioning owing to the participation in another area, or the situation when the performance of one role becomes better or easier due to the performance of another role. The positive affect perspective is based primarily on Sam D. Sieber's role accumulation theory [31] and Joän M. Patterson's concept of family resilience [33]. It counterbalances conflict by emphasising the social and psychological benefits of simultaneous participation in work and family life. It also emphasises that engagement in work and career does not exclude engagement in family life and motherhood. Indeed, it underpins the feeling of success that comes from balancing work and family roles.

Thus, the authors of this paper have asked the question: How do the above theoretical assumptions apply to the women who pursue a career in science and at the same time are mothers, actively raising children? The desk research carried out and the available empirical data indicate that analysing the situation of women scientists requires taking into account the bidirectional nature of the relationship between motherhood versus scientific career. On the one hand, the influence of the family is an important factor that shapes aspirations, opportunities, barriers, constraints and difficulties in career development. On the other hand, scientific work, with its individualised character, is very sensitive to social and family circumstances.

A scientific career is a specific type of career. It refers to the individual course of professional life, which concerns people striving to obtain academic degrees: doctor, habilitated doctor (post-doctoral degree), and its culmination is obtaining the academic title of professor (gaining it, however, does not end the scientific development). It is a process during which an individual accumulates scientific and research achievements, but also achievements in teaching and organisational work, necessary to obtain the indicated degrees and titles. Its perception depends on a person's emotional states, interpretation of work-related events, feelings about work experiences, aspirations, expectations, needs, sense of satisfaction and the person's value system [5].

Although the number of women in Europe pursuing scientific careers is slowly increasing, they are still significantly underrepresented and their potential is not fully recognized and valued. According to the She Figures 2018 report [34], women represent just over a third (33.4%)

| | 🇪🇺 | CZ | LT | PL | SK |
|---|---|---|---|---|---|
| PhD women graduates | 48.1 % | 43.7 % | 57.9 % | 56.3 % | 49.2 % |
| PhD women graduates<br>Information and Communication Technologies | 22.4 % | 4.05 % | 6.7 %<br>(1/6) | 10.2 % | 11.8 % |
| Women Researchers | 32.8 % | 26.6 % | 49.5 % | 38.1 % | 41.2 % |
| Women in grade A positions* | 26.2 % | : | 40.4 % | 25.2 % | 27.2 % |
| Women Heads of Higher Education institutions | 23.6 % | 10.3 %<br>(3/29) | 39 % | 19.6 % | 23 % |
| Women board leaders | 24.5 % | 0 %<br>(0/3) | 50 %<br>(2/4) | 19.4 % | 0 %<br>(0/7) |
| Women board members | 31.1 % | 16.7 % | 42.9 % | 24.9 % | 21.3 % |
| Publications with a gender dimension in their R&I content | 1.8 % | 1.76 % | 2.4 % | 2 % | 1.95 % |

*Equivalent to full-professorship positions

**Fig 2. Share of women in the science and research sector in selected EU countries (2021).** Source: She Figures 2021, after: A. Pépin, Webinar: "Gender equality in European Research, Innovation and Higher Education. How to enhance scientific excellence through Gender Equality Plans?", 10.03.2022.

of the total scientific population and their proportion in higher education institutions in Europe is only 22%. The latest data from the She Figures 2021 report [35] shows a similar percentage of 32.8% and a share of 23.6% in the leadership of higher education institutions (refers to the EU average).

Detailed data in relation to selected EU countries are graphically presented in Fig 2 below:

Data from the She Figures 2021 report also show that although the percentage of women holding the academic title of professor in the EU is gradually increasing, they constitute only slightly more than a quarter of the total number of its recipients, as presented in Fig 3.

Thus, in comparison to men, women are less likely to reach the highest stages of the scientific hierarchy in the academic sector (only Romania is balanced in this aspect and Latvia comes close).

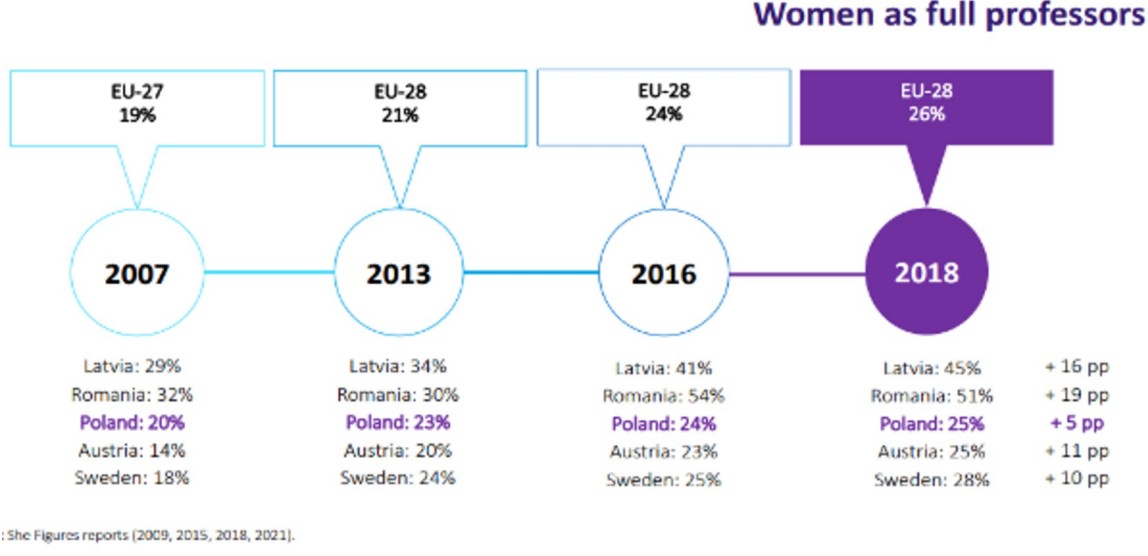

**Fig 3. Percentage of female professors in selected EU countries (2007, 2013, 2016, 2018).** Source: She Figures 2009, 2015, 2018, 2021, after: A. Knapińska, Women in science in Poland. Changing rules of the game, Webinar: "Gender equality in European Research, Innovation and Higher Education. How to enhance scientific excellence through Gender Equality Plans?", 10.03.2022.

**Table 1. Table scientific/research and research/research and teaching staff of academic institutions in Poland.** Source: compilation by E. Krause, based on the data provided to the author by OPI PIB (as of 29.01.2018).

| Scientific/research and research and teaching staff of academic institutions | MEN | | | | | | WOMEN | | | |
|---|---|---|---|---|---|---|---|---|---|---|
| Degree/ title | total | % total | number | % total | of 100% men: | % of those with the degree/title | number | % total | of 100% women: | % of those with the degree/title |
| | 70320, including | 100%, including | 39367, including | 56, including | | | 30953, including: | 44, including: | | |
| MA | 8354 | **11.9** | 4095 | **5.8** | **10.4** | **49** | 4259 | **6.1** | **13,8** | **51** |
| PhD | 35023 | **49.8** | 17361 | **24.7** | **44.1** | **49.6** | 17662 | **25.1** | **57,1** | **50.4** |
| PhD hab. | 17640 | **25.1** | 10910 | **15.5** | **27.7** | **61.8** | 6730 | **9.5** | **21,7** | **38.2** |
| Professor | 9303 | **13.2** | 7001 | **10** | **17.8** | **75.3** | 2302 | **3.3** | **7.4** | **24.7** |

In Poland, the situation of women in academia is comparable to that in other countries—their different political past or cultural differences are not relevant. Despite the political changes since the beginning of the 1990s, the transition from a centralized to a free market economy, the rapid massification of higher education, the increasing predominance of women among students, the percentage of women obtaining higher positions in the academic hierarchy is growing very slowly, although steadily. However, in Poland, three times fewer women reach higher levels of scientific careers than men (only 24.7% of people with the academic title of professor are women). The higher the scientific degree/title, the fewer women there are (National Information Processing Institute, National Research Institute 2018). The data has been presented in Table 1.

As statistics indicate, despite the gender balance in lower academic positions in Poland, when compared to men, three times fewer women reach higher academic career levels (only 24.7% of the individuals with the academic title of professor are women). One important reason for this disproportion is being a mother. The few studies to date have revealed that for the vast majority of mothers-scientists, motherhood is a factor that significantly delays their plans for a scientific career [36]. There is the so-called "hole in the pipeline" phenomenon, i.e. in science and higher education with the successive stages of scientific careers/levels in the scientific/academic hierarchy women "disappear"/gradually "leak out"—their number, in comparison to men, clearly decreases.

Studies on the division of responsibilities in families of scientists (in the USA and in European countries) indicate that most male professors have traditional families, in which it is primarily the female partner who takes care of the care and upbringing of children and taking care of the household. On the other hand, female professors much more often than their male colleagues do not have children or permanent partners [37–41]. The researchers emphasise that the burden of parental responsibilities is not limited to the stage of infancy or early child development. The number of hours devoted to childcare by female scientists averages 30 hours per week and remains more or less constant until the age of 50. The average number of hours worked by American scientists in independent positions is estimated at 50 hours per week. In the case of the University of California, Berkeley, it is as many as 57 hours. At the same institution, the total number of hours of work and parental and domestic responsibilities for female researchers aged 30–50 is as high as 100 hours per week (compared to 86 hours for men). Meeting the demands of a research career is therefore more difficult for women [42].

Other studies show that 80% of women lecturers at German universities are childless. For many German female academics, the decision to have a child is seen as an abandonment of their research career and a longer break to raise children is seen as a hindrance to their career development. They are convinced that these aspects cannot be reconciled [43].

Therefore, the high involvement of women in scientific work is associated either with child-lessness or with postponing the decision to procreate. Female physicists surveyed by Rachel Ivie were much more likely than male physicists to indicate that research work influenced their decision to start a family. For the aforementioned University of California, the age at which female scientists decide to have children is most often 38–40. Given the increasing time needed to find a stable job, delaying the decision to have children is becoming increasingly risky. Researchers from the cited University also declare that they have fewer children than they would like to have. The results of studies on professors working in the departments of astronomy, physics and biology are similar: women have fewer children than men in the same positions and would also like to have more children than they have [44]. It can therefore be emphasised, as Donna K. Ginther and Shulamit Kahn fund out, that although gender does not directly discriminate against women in science, motherhood does [45].

Starting one's own family and being a parent can be a constraint or obstacle to women's scientific careers. This has been confirmed by further studies, including those by Cornelia Lawson and Aldo Geuna. According to the researchers, the results are universal for all fields of science and for most countries—female scientists with young children received less funding to conduct their own research and also obtained lower citation rates than their male colleagues and childless colleagues in similar positions. The authors of the study emphasise that mothers-scientists (of young children) find it more difficult to obtain support for their projects or access to research funding than male scientists [46].

In conclusion, as the analysis of the literature indicates, for the vast majority of mothers of female scientists, motherhood is a factor that significantly delays their career development plans and even causes women to leave or slow down their scientific careers [2]; decisions on procreation are often postponed until years after the completion of the doctorate or habilitation (this means late motherhood with its risk of adverse consequences [3]); women who are successful in science (in an objective sense) are more likely to be single and childless, which allows them to focus on their professional life and scientific development [47].

From the data outlined above, it is clear that the costs of reconciling a career in science and a parental role are high for women. They experience many difficulties in balancing maternity responsibilities and research.

## The mechanisms delaying women's careers

It is also worth pointing out the main mechanisms of delays in the development of women's scientific careers, which may provide an interpretative background for the causes of gender imbalance in achieving successive stages of scientific careers. The literature on the subject points to many causes of gender imbalance in science and their different types, e.g. non-institutional and institutional, among which one can distinguish biological and cultural differences or historical background [48, 49]. The situation of female researchers in the science and higher education sector can be analysed in the context of mechanisms determining (limiting, hindering) women's professional and scientific careers. They are related to such phenomena as

- glass ceiling [50, 51]—it is a phenomenon describing invisible obstacles standing in the way of women wishing to advance, and thus hindering their professional careers and associated with the occurrence of professional discrimination of vertical nature. Women are unable to climb the career ladder and reach the highest levels of management

- glass cliff [51, 52]—a phenomenon denoting situations in which a woman's work in a "masculine" (high, decision-making) position (high, decision-making) position is unstable, difficult to maintain and exposed to constant criticism

- glass escalator effect [51, 53]—a discriminatory mechanism in the system of science and higher education, which refers to an invisible force that elevates men to higher levels of professional and scientific careers. It functions on the basis of a stereotypical belief that men (as a group) take decisions and positions that require it more easily and are more suitable for them than for women

- the runaway ladder [49]—this term can be associated with a lack of solidarity among women. Those who are determined and strong pursue their careers through individual strategies. Men, on the other hand, help, support and employ each other. For example, a female scientist may be refused employment or promotion by a female superior or a woman of prominence

- glass walls [49, 51]—another type of horizontal occupational discrimination, which means that women do not have access to prestigious, well-paid jobs, industries or positions, and therefore find employment in auxiliary sectors

- sticky floor [54]—this phenomenon represents situations in which women are "stuck" to their professions and positions—they dominate in professions with low prestige and low income, where there are no opportunities for advancement. In relation to female scientists, it means that more often than their male colleagues they are involved in teaching, which is perceived as less prestigious than conducting research

- velvet ghetto [55]—a term used to describe the fact that women are hindered or precisely prevented from holding managerial and decision-making positions, including positions related to technical sciences, production, marketing. It is also characterized by disrespectful treatment of the women holding managerial positions in industries or professions traditionally assigned to men.

These phenomena, present in academic circles, do not facilitate professional advancement of women in the sector of science and higher education. And as it has already been pointed out in the article—the higher the level of scientific career, the fewer women there are. Few of them also reach the highest positions in the professional hierarchy. Thus, women disappear somewhere "along the way" at different stages of scientific career. It is helpful here to describe further phenomena such as the "magic vanish box" and the leaky pipeline. These terms are related to each other. "Magic vanish box [56]" is in fact the term used to describe the "disappearance" of women from the scientific world and their entry (at that time) into alternative career paths. Women's talents "leak out" as they move up the scientific career ladder. In this way, women's talents are wasted, yet qualifications and competencies have no gender. Wasting their potential is particularly noticeable in the sphere of science and higher education. And talented female scientists gradually leak out through the "leaky pipeline [57, 58]", limiting or completely suspending their scientific and professional development. They "drop out" from higher education institutions and devote themselves to activities other than a scientific career. Family life, especially motherhood, is largely responsible for this "drop out" or "leakage". Most female scientists leave the science and higher education sector between the ages of 30–39. This is because this is the period when they are (usually) heavily involved in caring for and raising children. Previous research has revealed that for the vast majority of scientist mothers, motherhood is a factor that delays their plans for a career in science (As indicated by the study of Ewa Krause; the full results yet to be published).

## Data mining and basket analysis

In this paper, the results of the study have been analysed twice; the other time it was done using one of the data mining techniques. Data mining is the process of obtaining, extracting

unstructured information from very large data sets. Intelligent methods are often used to extract patterns and find order in historical data. One type of data mining is association rule mining. It is used to discover association relationships between elements belonging to large data sets. The basic association rule is X → Y; i.e., if X is true for an instance in the dataset, then Y is also true for it. Here, X is called an antecedent and Y is called a consequent. Antecedents are the elements that come first, and all the consequents follow them. There are three indicators used to assess how significant the relationship is. They are as follows: support, confidence and lift. Support determines the level of popularity of a given combination of elements in a data set. It is the ratio of the number of occurrences of a set of items X and Y set against the total number of items in the set. In other words, if out of 100 items, the item X occurs 20 times, the support will be 20/100, 0.2 or 20%. Support is expressed by the Formula: $Support = \frac{freq(X,Y)}{N}$ [59]. Confidence is a measure of the probability of an item Y occurring together with an item X; it is expressed as the proportion of the number of times X and Y occur together to the number of times X occurs. The Formula is: $Confidence = \frac{freq(X,Y)}{freq(X)}$ [59]. Since this measure may misrepresent the significance of association (e.g., in cases when item Y is also popular), the third measure (lift) is used. Lift determines how likely it is that item Y will appear together with item X, given the popularity of item Y. Lift's Formula is: $Lift = \frac{Support}{Support(X)*Support(Y)}$ [60]. So, lift is a measure that determines how strong an association is. If lift is greater than 1, it means that the occurrence of X does lead to Y. The higher the lift, the stronger the association. If lift is close to 1, it indicates that the elements do not affect one another other. Finally, if lift is lower than 1, then the item X has a negative effect on the appearance of the element Y [61].

## Materials and methods

Besides the basket analysis, the following methods have also been employed in the course of the experiments presented in this paper:

• Method of critical analysis of sources (literature) [62].

  This method allows one to establish a list of the most important items of specialized or methodological literature related to a large extent to the problem. It is a common method and is applicable to all fields of scientific inquiry. Its essence is to demonstrate the expediency, originality and new treatment of the problem that has emerged and undertaken for research. It is necessary to demonstrate by this method what we know and what we do not know, what already exists and is contained (known) in the literature, and what does not exist and needs to be known and proven by research. Cognition is carried out by means of analysis and criticism of the literature of the subject (issue) under study. This method involves studying the news recorded in studies, documents and other communications. The selection of the sample, that is, the fragments of the messages analyzed, depends on what is the unit of analysis. These can be, for example, specific events, characters, individual words, definitions, expressions and others. Content analysis is carried out in three stages: coding of the material (both explicit and implicit content), counting the coded by category of content, comparative analysis of the collected empirical material.

• Formal-legal document analysis method [62, 63]

  This method is widely spread in the social sciences, it is applicable where the study takes into account the most diverse documents. The documents to be studied can be, therefore, any human products (material objects) that express a thought, vision, mission, achievement,

proposal and serve to reproduce the actual activity or state of the organizational structure under study in the form of a legally or even customary document. It is, therefore, an extremely important, extensive, detailed and specific source material (for example, terms of reference) concerning a particular institution or person. In general, a distinction is made between classical (qualitative or descriptive) and modern (quantitative) document analysis. Classical document analysis is mainly based on their historical and literary interpretation. It is a search for individual characteristics specific to the analyzed document and its creator. Modern document analysis is an attempt to overcome the subjective nature of traditional (classical) analysis. It consists primarily of quantitative description and analysis of documents, with such description and analysis not limited to the use of absolute numbers or percentages. Great importance is attached to the accurate determination of the cognitive value of documents, including, in particular, the confirmation of their reliability and authenticity, that is, it is sought to demonstrate that the documents included in the analysis can be a legitimate basis for solving the problem of interest to the researcher, and that the time of their creation, author or creator and place of origin are well known to them. To each of these document analysis, different varieties of documents are subordinated. Content analysis involves the interpretation of the contents contained in the documents. Formal analysis of documents, on the other hand, is concerned with the external description of their appearance, the way they were drawn up, the degree of permanence or relevance with the intention that guided or was intended in the course of their creation.

- Diagnostic survey method [63, 64]

    This method is sometimes—for the purposes of pedagogical research—called a diagnostic survey, representative group survey, opinion survey or simply a survey. In the most general terms, it is understood as a method of research, the primary function of which is to collect information about the problems of interest to the researcher as a result of verbal reports of the subjects, called respondents. The constructive feature of the survey method is "questioning," that is, probing opinions. In the case of written responses, it takes the form of surveys, and in the case of oral responses, it takes the form of interviews.

- Statistical methods [62, 63]

    These are ways of quantitatively describing mass phenomena and the relationships between them, as well as presenting the results in a conventional form (tables, graphs, average values, correlation). From the results obtained, conclusions are derived on the basis of probability theory.

## Research questions

From the literature review presented outlined above, it is clear that the costs of reconciling a career in science and a parental role are high for women. They experience many difficulties in balancing maternity responsibilities and research. Taking into account the above background, as well as research results from other countries described in the literature, the authors of this article posed the following three research questions:

- Research question 1. Do the surveyed women employed at academic institutions in Poland experience conflict between scientific career and motherhood? If so, what kind of conflict?

- Research question 2. What are the dimensions of the conflict between academic career and motherhood, according to the surveyed women employed at academic institutions in Poland?

- Research question 3. Is the basket analysis method able to find possible correlations between the factors characterising the respondents and the types and dimensions of the conflict between academic career and motherhood they claim to experience?

## Study design, setting and course

The course of the study has been shown in the pipeline in Fig 4.

**Study group.**  The criteria for selecting participants for the study and representativeness of the research sample were:

- gender of the study subjects—women only;

- being a mother;

- possessing specific professional titles/degrees/academic title (MA, PhD, PhD Hab., Professor);

- being employed in a position that obliges them to conduct research and development activity at an academic institution in Poland (scientific/research and scientific/research/didactic positions: assistant, assistant professor, associate professor, full professor, and others);

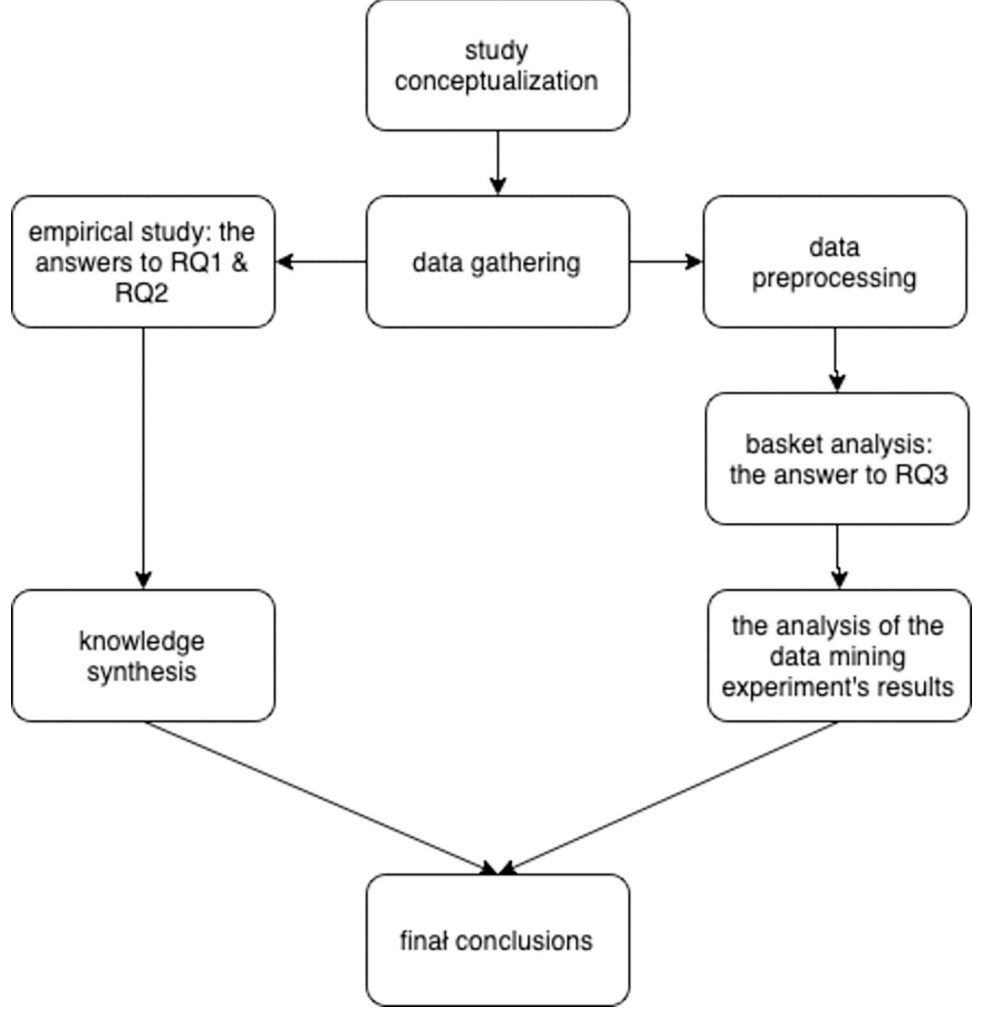

**Fig 4. The course of the experiments presented in this paper.**

- willingness to participate in the research;

- consent of the Ethics Committee of Kazimierz Wielki University in Bydgoszcz.

The written invitation to the research received a positive opinion of the Faculty Committee for Research Ethics of the Faculty of Pedagogy at the University of Bydgoszcz and was sent to all academic universities in Poland (e-mails with the invitation were sent to university authorities—especially to offices, secretariats and directly to Rectors and Vice-Rectors for Science). In response, e-mail and telephone consents to conduct the survey were obtained. A link was sent through the universities to the female researchers employed at them.

These universities were the primary place of work for the majority of women scientists//researchers in Poland and it is only in these types of universities that one can pursue degrees: Ph.D., D.Sc., and which fall under the term "scientific career". Therefore, the established research area were the academic universities in the country. In Poland, according to the list of the Information Processing Center—National Research Institute, in 2018 there were 122 academic universities and the invitation was sent to all of them.

A random-quota selection of the research sample was applied. The research itself was of national scope and was designed and conducted by Ewa Krause, using the statistical survey method and questionnaire technique.

The relevance and reliability of the questionnaire administered to the surveyed female scientists was optimized by conducting a pilot study (traditional version: February—April 2019; online: November—December 2019). The actual study was conducted in January, February and March 2020. The experiment was conducted according to the proper methodology, taking into account the control, replication and sample size.

**Ethics approval and consent to participate.** Participation in the study was fully voluntary, informed and consensual. The data obtained from the respondents were anonymised, which made it impossible to identify specific university employees. Thus, the Ethical Committee of Kazimierz Wielki University agreed to collect the data and conduct all subsequent research on the obtained data set.

## Results

### Characteristics of the participants

Some characteristics of the researcher mothers are presented in Tables 2–7 (More than half of the female scientists (53.3%) fulfilling the role of mother have two children and almost 1/3 (32.2%) have an only child. Much fewer (13.6%) have three children. One respondent has four children and one respondent has five children–the latter is bringing up foster children. The average number of children had by the surveyed mothers is 1.83).

### Part 1—empirical research. Survey results

**The answer to the research question no. 1.** Moving on to the results part, an answer was firstly sought to the research question: Do the surveyed women employed at academic institutions in Poland experience any conflict between scientific career and motherhood? If so, what kind of conflict? The empirical data obtained in the course of the research are shown in Table 8.

Most respondents declare that they combine their maternal role with their professional role (female scientist), but"sometimes" feel that these roles are in conflict (45.8%). A significant proportion of female respondents indicate that they experience this conflict"often" (28.4%). Experiencing constant, permanent conflict was reported by the smallest percentage of female respondents (7.5%). When combining the answers regarding experiencing any type of conflict,

**Table 2. Characteristics of the researched women on the basis of their professional title/degree/academic title.**
N = 334. Source: Study by Ewa Krause.

| No. | Professional title/ academic degree/ academic title | The number of answers | Percent |
|---|---|---|---|
| 1. | MA | 20 | 6.0 |
| 2. | PhD | 189 | 56.6 |
| 3. | PhD Hab. | 102 | 30.5 |
| 4. | Professor | 23 | 6.9 |

**Table 3. Characteristics of the surveyed women by age.** N = 334. Source: Study by Ewa Krause.

| No. | Age | Number of answers | Percent |
|---|---|---|---|
| 1. | 26–30 | 4 | 1.2 |
| 2. | 31–35 | 53 | 15.9 |
| 3. | 36–40 | 71 | 21.3 |
| 4. | 41–45 | 90 | 26.8 |
| 5. | 46–50 | 55 | 16.5 |
| 6. | 51–55 | 26 | 7.8 |
| 7. | 56–60 | 13 | 3.9 |
| 8. | 61–65 | 11 | 3.3 |
| 9. | 66 or older | 11 | 3.3 |

**Table 4. Characteristics of the surveyed women by marital status.** N = 333 (one woman did not answer the question). Source: Study by Ewa Krause.

| No. | Marital status | Number of answers | Percent |
|---|---|---|---|
| 1. | in a formal relationship | 293 | 88 |
| 2. | in an informal relationship | 18 | 5.4 |
| 3. | single | 22 | 6.6 |
| 4. | total | 333 | 100 |

**Table 5. Characteristics of the surveyed women with regard to their experience of divorce.** N = 334 (no data available for 1 mother). Source: Study by Ewa Krause.

| No. | Divorced /not divorced | Number of answers | Percent |
|---|---|---|---|
| 1. | Divorced | 35 | 10.5 |
| 2. | Not divorced | 298 | 89.5 |
| 3. | Total | 333 | 100 |

**Table 6. Characteristics of the surveyed women with regard to the assessment of their economic situation.**
N = 334. Source: Study by Ewa Krause.

| No. | Assessment of economic situation | Number of answers | Percent |
|---|---|---|---|
| 1. | Definitely bad/ very bad | 1 | 0.3 |
| 2. | Rather bad | 18 | 5.4 |
| 3. | I do not know | 12 | 3.6 |
| 4. | Rather good | 230 | 68.8 |
| 5. | Definitely good/ very good | 73 | 21.9 |

**Table 7. Number of children the studied mothers have.** N = 334 (no data for 2 mothers). Source: Study by Ewa Krause.

| No. | Number of children | Number of answers | Percent |
|---|---|---|---|
| 1. | 1 child | 107 | 32.2 |
| 2. | 2 children | 178 | 53.6 |
| 3. | 3 children | 45 | 13.6 |
| 4. | 4 children | 1 | 0.3 |
| 5. | 5 children | 1 | 0.3 |
| 6. | Total | 332 | 100 |

**Table 8. Conflict between scientific career and motherhood as perceived by mothers-scientists.** Source: Study by Ewa Krause.

| No. | Answer | Number of answers | Percent |
|---|---|---|---|
| 1. | I definitely manage to reconcile my maternal role with my professional role (female scientist). | 59 | 17.7 |
| 2. | I combine my maternal role with my professional role (as a female scientist), but sometimes I experience conflict between these roles | 153 | 45.8 |
| 3. | I combine my maternal role with my professional role (female scientist), but I often experience conflict between these roles. | 95 | 28.4 |
| 4. | My roles (maternal and professional/ female scientist) are in constant conflict. | 25 | 7.5 |
| 5. | Difficult to say/don't know | - | - |
| 6. | Other | 2 | 0.6 |
| 7. | Total | 334 | 100 |

it turns out that the vast majority of respondents experience it (81.7%). Only 17.7% of the respondents declare that they definitely manage to reconcile their maternal role with their professional role. None of the respondents chose the option"difficult to say/don't know".

According to the surveyed women, female scientists in their role as mothers face the lack of understanding and lack of recognition by the professional scientific community of the interruptions in their scientific activity resulting from motherhood. Motherhood means a decrease in availability and scientific mobility, which are required by the environment. This claim has been confirmed by the following views, expressed by the surveyed women:

- "For the manager, a woman scientist having young children means frequent absences from work (taking care of a sick child) and the inability to travel for a longer period of time (lack of scientific mobility)" [PhD, medical university, field of medical sciences, 41 years old, 2 children];

- "Decreased availability (which is necessary for certain laboratory work)" [Ph.D., university, science field, 39 years old, 1 child];

- "It is unfair to require full availability from any employee and call them at 4am for business. I think this is not just about the treatment of women mothers, but the generally sick, feudal system at universities" [Ph.D., university of economics, field of social sciences, 45 years old, 2 children];

- "Certainly, women who give birth to children are visibly delayed in reaching the next stages of their academic careers; the mothers also have neither professional nor social support in the workplace, but they can pursue their careers—for that they are not expected to perform at their best there" [Ph.D., university, humanities field, 49 years old, 2 children];

The following views emphasize the fact that the realization of an academic career is always done at someone's expense, most often at the expense of family life and one's children. Here are some sample statements on this matter, made by the surveyed women:

- "Career in our environment is always at the expense of family, children or own health (as in my case) (. . .)" [Ph.D., university, field of science and life sciences, 49 years old, 2 children];

- "Even if a woman manages to advance and develop academically and professionally, it is at the expense of her family, making great sacrifices, working beyond her physical and mental capabilities" [PhD, medical school, field of medical and health sciences, 39 years old, 2 children].

The results relating to the parental/professional situation of mothers-scientists turned out to be statistically significant due to the fact that the respondents held a specific professional or scientific degree/title (Chi^2 Pearson p = 0.00042; Chi^2 NW p = 0.00121).

Considering the results in relation to the individual groups of female researchers, it may be noted that the higher a given woman is in the scientific hierarchy in terms of degree/title, the more often she declares that she definitely manages to reconcile her maternal role with her professional role–this refers to 47.8% of the respondents with the academic title of professor; 22.5% of the respondents with the academic degree of"dr habilitowany/associate professor/ScD"; 12.7% of PhDs (12.7%) and one woman with the professional title of magister/Master's (MA) (5.0%). The differences are therefore very significant—between a Master's degree and a Professor's degree by as much as 42.8 pp. There is also not a single female professor indicating that:"my roles (maternal and professional/women scientist) are in constant conflict". Thus, the lower the researcher is in this hierarchy, the more frequent is the experience of mother-scientist conflict (Master—35.0%; Doctor– 32.08%; Postdoctoral -22.5%; Professor– 13.0%). The results are shown in Table 9.

**The answer to the research question no. 2.**    Next, the answer to the question was sought: what are the dimensions of the conflict between scientific career and motherhood in the opinion of the surveyed women employed in the academic institutions?

The time dimension of this conflict was indicated by the majority of the respondents–a total of 79.3%, including "definitely experienced" or "experienced" by almost half of the respondents (45.8%). The data are shown in Table 10.

**Table 9. Conflict of scientific career vs. motherhood in the opinion of mothers of female scientists by professional title/degree/title.** Source: Study by E. Krause.

| Professional title/ degree/ academic title | Defining one's own parental/ professional situation | | | | | |
|---|---|---|---|---|---|---|
| | My roles are in constant conflict | I often experience role conflict | I sometimes experience role conflict | I definitely manage to reconcile my roles | Other | Total answers and % |
| **MA—number of answers** | 2 | 7 | 10 | 1 | - | 20 |
| % of MAs | 10.0 | 35.0 | 50.0 | 5.0 | - | 100 |
| % of total | 0.6 | 2.1 | 3.0 | 0.3 | - | 6.0 |
| **PhD—number of answers** | 19 | 62 | 83 | 24 | 1 | 189 |
| % of PhDs | 10.1 | 32.8 | 34.9 | 12.7 | 0.5 | 100 56.6 |
| % of total | 5.7 | 18.6 | 24.8 | 7.2 | 0.3 | |
| **PhD Hab.- number of answers** | 4 | 23 | 52 | 23 | - | 102 |
| % of PhD Habs. | 3.9 | 22.5 | 51.1 | 22.5 | - | 100 |
| % of total | 1.2 | 6.9 | 15.5 | 6.9 | - | 30.5 |
| **Professor—number of answers** | - | 3 | 8 | 11 | 1 | 23 |
| % of Professors | - | 13.0 | 34.8 | 47.8 | 4.4 | 100 |
| % of total | - | 0.9 | 2.4 | 3.3 | 0.3 | 6.9 |

**Table 10. Time-related dimension of conflict between the roles of mother and scientist.**

| No. | Answer | Number of answers | % |
|---|---|---|---|
| **1.** | Definitely yes | 153 | 45.8 |
| **2.** | Rather yes | 112 | 33.5 |
| **3.** | Hard to tell | 25 | 7.5 |
| **4.** | Rather not | 28 | 8.4 |
| **5.** | Definitely not | 16 | 4.8 |
| **6.** | Total | 334 | 100 |

The results in terms of the time dimension turned out to be statistically significant due to the fact that the researched female scientists hold a specific professional or scientific degree/title (Chi^ 2Pearson p = 0.00102; Chi^ 2NW p = 0.00102). The results relating to these relationships are presented in Table 11.

The results shown above indicate that the lower a given woman is in the scientific hierarchy in terms of degree/title, the more often she declares to feel the time dimension of the mother-scientist role conflict. This applies to almost all female researchers with a master's degree (95.0%, of which the answer"definitely yes" was chosen by 50.0% of the respondents, and the answer"rather yes"– 45.0%); the majority of PhDs (86.08%, of which 51.3% chose the answer"-definitely yes" and 35.5% the answer"rather yes" - 35.5%); the majority of post-doctoral fellows (68.06%, of which 39.2% chose"definitely yes" and 29.4%"rather yes") and slightly more than half of the respondents with a professor's degree (52.2%; 26.1% of respondents for each of the indicated answers). Here, the differences are great, too–between the professional title of Master and that of Professor by 42.8 percentage points. There is also not a single female Master not feeling this dimension of the role conflict.

The financial dimension of the mother-scientist role conflict is indicated by only half of the respondents–it concerns or concerned 49.6% of them, 27.2% of which chose the answer"definitely yes" and 22.4% the answer"rather yes". This data is shown in Table 12.

The emotional dimension of the conflict between the roles of mother and scientist is indicated by the vast majority of respondents–it concerns or concerned 72.7%, of which"definitely yes" was selected by 38.6% and"rather yes" by 34.1%. A minority of respondents claim that this dimension is not experienced (18.0% in total). The remaining respondents did not answer this question (9.3%). The results are shown in Table 13.

**Table 11. Time-related dimension of the mother-scientist role conflict by professional title/degree/academic title.** Source: Study by Ewa Krause.

| Professional title/ degree/ academic title | Time-related dimension of the mother-scientist role conflict | | | | | |
|---|---|---|---|---|---|---|
| | definitely not | rather not | hard to tell | rather yes | definitely yes | total and % |
| **MA–number of answers** | - | - | 1 | 9 | 10 | 20 |
| % of MAs | - | - | 5.0 | 45.0 | 50.0 | 100 |
| % of total | - | - | 0.3 | 2.7 | 3.0 | 6.0 |
| **PhD–number of answers** | 8 | 11 | 6 | 67 | 97 | 189 |
| % of PhDs | 4.2 | 5.8 | 3.2 | 35.5 | 51.3 | 100 |
| % of total | 2.4 | 3.3 | 1.8 | 20.1 | 29.0 | 56.6 |
| **PhD Hab.- number of answers** | 5 | 12 | 15 | 30 | 40 | 102 |
| % of PhD Habs. | 4.9 | 11.8 | 14.7 | 29.4 | 39.2 | 100 |
| % of total | 1.5 | 3.6 | 4.5 | 9.0 | 12.0 | 30.5 |
| **Professor—number of answers** | 3 | 5 | 3 | 6 | 6 | 23 |
| % of Professors | 13.0 | 21.8 | 13.0 | 26.1 | 26.1 | 100 |
| % of total | 0.9 | 1.5 | 0.9 | 1.8 | 1.8 | 6.9 |

**Table 12. The financial dimension of the mother-scientist role conflict.** Source: Study by Ewa Krause.

| No. | Answer | Number of answers | % |
|---|---|---|---|
| 1. | Definitely yes | 91 | 27.2 |
| 2. | Rather yes | 75 | 22.4 |
| 3. | Hard to tell | 61 | 18.3 |
| 4. | Rather not | 74 | 22.2 |
| 5. | Definitely not | 33 | 9.9 |
| 6. | Total | 334 | 100 |

It is worth adding that the situation of mothers-scientists depends on the resources enabling the reconciliation of parental and professional roles. The assessment of particular resources is shown in the following statistics:

1. assessment of partner's support in child/children care as a resource enabling reconciliation of parental and professional roles by surveyed mothers-scientists: highly positive– 47.0%; positive– 24.8%; average– 14.1%; negative– 6.1%; highly negative– 7.8%.

2. assessment of the availability of assistance from third parties as a resource for reconciling parental and professional roles: highly positive– 17.4%; positive—30.8%; average– 25.7%; negative– 17.1%; highly negative– 9.0%.

3. assessment of own family financial resources in the context of reconciling parental and professional roles: highly positive– 19.5%; positive– 32.0%; average– 40.4%; negative– 5.7%; highly negative– 2.4%.

4. assessment of the availability of assistance from the work environment as a resource for reconciling parental and professional roles: highly positive– 4.5%; positive– 11.7%; average 44.6%; negative– 24.2%; highly negative– 15.0%.

5. evaluation of availability of institutional solutions as a resource enabling reconciliation of parental and professional roles: highly positive– 2.1%; positive -9.3%; average– 41.9%; negative– 28.7%; highly negative– 18.0%.

Summarizing the results concerning the assessment of the resources enabling the reconciliation of parental and professional roles by the surveyed mothers, it should be noted that the highest rated resource (i.e., assessed either as positive or highly positive) is the partner's support in caring for the child/children (71.8%); followed by the family's own finances (51.5%) and the availability of help from third parties (48.2%). The availability of help from the work environment (16.2%) and institutional arrangements (11.4%) are much less well perceived. Additionally, personal resources, appropriate skills and character traits conducive to role reconciliation are also highlighted in the researchers' responses.

**Table 13. The emotional dimension of the mother-scientist role conflict.** Source: Study by Ewa Krause.

| No. | Answer | Number of answers | % |
|---|---|---|---|
| 1. | Definitely yes | 129 | 38.6 |
| 2. | Rather yes | 114 | 34.1 |
| 3. | Hard to tell | 31 | 9.3 |
| 4. | Rather not | 43 | 12.9 |
| 5. | Definitely not | 17 | 5.1 |
| 6. | Total | 334 | 100 |

## Part 2—basket analysis of the obtained data

**Experimental setup.**   Having gathered, analysed and interpreted the data from the study, the authors went on to apply the basket analysis principles to it, and analyse it according to the methodology. The data from the empirical study were pre-processed by extracting only the information on female researchers which was relevant, and adapting the dataset to Python format. The dataset is publicly available [65]. For the purpose of this study, the apriori algorithm was selected. It is one of the most popular association rule mining algorithms. It was developed by Agrawal [66]; later, it was scrutinized by Bhargava and Selwal [67]. The underlying assumption of the algorithm is that all the subsets of a frequent set must be frequent, too. On the other hand, if a set is not frequent, then its subsets will not be frequent either. When working with the apriori algorithm, it is advisable for the person applying the algorithm to personalise the thresholds/frequency levels in the course of the experiment, according to their experience and current needs, etc. The apriori algorithm's most popular application is the so-called market basket analysis—the study of transactions aimed at finding out which items are purchased together. The algorithm lets one uncover interesting, surprising associations within vast data sets. The advantages of the algorithm are that it is friendly and easy to use. The main disadvantages consist in the fact that the algorithm needs a lot of resources and computational time, especially if it is used to examine big amounts of data, or if the minimum thresholds are set to very low. The reason for choosing this particular algorithm was the desire to find different, unusual associations and relationships than those that intuitively seem true to human researchers, or to confirm the previously drawn conclusions were valid. Additionally, the apriori algorithm is most often used in business and commerce (see [59, 68]), there have also been numerous other applications of basket analysis in other fields, e.g., for analysing medical data [69]. However, to the best of the Authors' knowledge, the subject literature has yet to record the application of the apriori algorithm for data mining within such an extensive social sciences survey, let alone one concerning women scientists.

The measure thresholds used in the following study, determined by experimentation and finetuned according to the Authors' expertise, are: minimum support—0.05 (5%), minimum confidence—0.3 (30%), minimum lift—2. Using such thresholds, the algorithm found 28 association rules among the data. Below, they are arranged according to the lift measure, from the highest to the lowest. Thus, by assumption, the associations are arranged from the strongest to the weakest ones. It should be noted that due to the assumption of a minimum lift = 2, all listed associations are significant. The apriori association rules found by the algorithm should be interpreted as follows:

1. If a female respondent indicated the answer"financial conflict–definitely not/strongly disagree", she also answered:"economic situation–definitely good/very good", and:"I am not single", support: 0.06, confidence: 0.58, lift: 2.66.

2. If the female respondent indicated the answer"time conflict—definitely yes/strongly agree", she also answered:"age 36–40","PhD title","I am not single" and"I combine my maternal role with my professional role (female scientist), but sometimes I experience the conflict of these roles", support: 0.05, confidence: 0.47, lift: 2.63.

3. If the female respondent indicated the answer"financial conflict: definitely not/strongly disagree", she also answered:"economic situation—definitely good/very good" support: 0.06, confidence: 0.58, lift: 2.63.

4. If the female respondent indicated the answer"time conflict–definitely yes/strongly agree", she also answered:"age 36–40","PhD title" and"I combine my maternal role with my

professional role (female scientist), but sometimes I experience the conflict of these roles" and:"I am not single", support: 0.05, confidence: 0.47, lift: 2.54.

5. If the female respondent indicated the answer"time conflict—definitely yes/strongly agree", she also answered:"financial conflict—definitely yes/strongly agree","I combine my maternal role with my professional role (woman scientist), but sometimes I experience the conflict of these roles","number of children: 2", support: 0.06, confidence: 0.34, lift: 2.3.

6. If the female respondent indicated the answer"age—31–35", she also answered:"doctoral degree","number of children—1","I am not single", support: 0.06, confidence: 0.38, lift: 2.28.

7. If the female respondent indicated the answer"I definitely manage to reconcile my maternal role with my professional role (female scientist)", she also answered:"economic situation—rather good","time conflict—definitely yes/strongly agree", support: 0.05, confidence: 0.68, lift: 2.26.

8. If the female respondent indicated the answer"time conflict–definitely yes/strongly agree", she also answered:"age 36–40","economic situation–rather good" and"I combine my maternal role with my professional role (woman scientist), but sometimes I experience the conflict of these roles" and:"I am not single", support: 0.05, confidence: 0.45, lift: 2.22.

9. If the female respondent indicated the answer"age 36–40", she also answered:"economic situation—rather good" and"I combine maternal role with professional role (woman scientist), but sometimes I experience the conflict of these roles","support: 0.05, confidence: 0.45, lift: 2.22.

10. If the female respondent indicated the answer"PhD title", she also answered:"I combine maternal role with professional role (woman scientist), but sometimes I experience the conflict of these roles","economic situation—rather good","I am not single","age 36–40","support: 0.06, confidence: 0.34, lift: 2.21.

11. If the female respondent indicated the answer"time conflict–definitely yes/strongly agree", she also answered:"financial conflict—definitely yes/strongly agree","I am not single","I combine maternal role with professional role (woman scientist), but sometimes I experience the conflict of these roles","number of children—2", support: 0.06, confidence: 0.31, lift: 2.17.

12. If the female respondent indicated the answer"time conflict–definitely yes/strongly agree", she also answered:"age 36–40","I am not single","I combine maternal role with professional role (female scientist), but sometimes I experience the conflict of these roles", support: 0.07, confidence: 0.58, lift: 2.14.

13. If the female respondent indicated the answer"PhD title", she also answered:"I combine maternal role with professional role (woman scientist), but sometimes I experience the conflict of these roles","economic situation—rather good","age 36–40","support: 0.06, confidence: 0.34, lift: 2.13.

14. If the female respondent indicated the answer"time conflict—definitely yes/strongly agree", she also answered:"age 36–40","I combine my maternal role with my professional role (woman scientist), but sometimes I experience the conflict of these roles", support: 0.07, confidence: 0.61, lift: 2.12.

15. If the female respondent indicated the answer"PhD title", she also answered:"My roles (maternal and professional/women scientist) are in constant conflict","economic situation

—rather good","number of children—2","I am not single","time conflict—probably yes/ rather agree", support: 0.06, confidence: 0.67, lift: 2.09.

16. If the woman surveyed indicated the answer"number of children—1", she also answered:"I combine my maternal role with my professional role (woman scientist), but sometimes I experience the conflict of these roles","PhD title","time conflict—definitely yes/strongly agree", support: 0.05, confidence: 0.61, lift: 2.08.

17. If the female respondent indicated the answer"economic situation—rather good", she also answered:"financial conflict—definitely yes/strongly agree","I am not single","I combine my maternal role with my professional role (woman scientist), but sometimes I experience the conflict of these roles", support: 0.06, confidence: 0.43, lift: 2.07.

18. If the female respondent indicated the answer"financial conflict–definitely yes/strongly agree", she also answered:"I am not single","number of children—2","I combine maternal role with professional role (woman scientist), but sometimes I experience the conflict of these roles", support: 0.06, confidence: 0.59, lift: 2.06.

19. if the female respondent indicated the answer"age—31–35", she also answered:"doctoral degree","number of children—1", support: 0.06, confidence: 0.38, lift: 2.06.

20. If the female respondent indicated the answer"time conflict–definitely yes/strongly agree", she also answered:"age 41–45","financial conflict–definitely yes/strongly agree", support: 0.06, confidence: 0.46, lift: 2.06.

21. If the female respondent indicated the answer"PhD title", she also answered:"financial conflict—definitely yes/strongly agree","time conflict–definitely yes/strongly agree","I am not single","I combine maternal role with professional role (woman scientist), but sometimes I experience the conflict of these roles", support: 0.06, confidence: 0.43, lift: 2.05.

22. If the female respondent indicated the answer"time conflict–definitely yes/strongly agree", she also answered:"age 41–45","financial conflict–definitely yes/strongly agree","I am not single" support: 0.05, confidence: 0.41, lift: 2.03.

23. If the female respondent indicated the answer"time conflict–definitely yes/strongly agree", she also answered:"number of children—2","I combine the maternal role with the professional role (woman scientist), but sometimes I experience the conflict of these roles", support: 0.07, confidence: 0.58, lift: 2.03.

24. If the female respondent indicated the answer"economic situation—rather good", she also answered:"financial conflict—definitely yes/strongly agree","time conflict—definitely yes/ strongly agree","I combine maternal role with professional role (woman scientist), but sometimes I experience the conflict of these roles", support: 0.06, confidence: 0.46, lift: 2.03.

25. If the female respondent indicated the answer"doctoral title", she also answered:"financial conflict—definitely yes/strongly agree","economic situation–rather good","time conflict— definitely yes/strongly agree","I am not single", support: 0.07, confidence: 0.85, lift: 2.03.

26. If the female respondent indicated the answer"time conflict–definitely yes/strongly agree", she also answered:"I combine my maternal role with my professional role (woman scientist), but sometimes I experience the conflict of these roles","financial conflict—definitely yes/strongly agree","I am not single", support: 0.09, confidence: 0.41, lift: 2.02.

27. If the female respondent indicated the answer"doctoral degree", she also answered:"financial conflict—definitely yes/strongly agree","time conflict—definitely yes/strongly

agree","I combine my maternal role with my professional role (woman scientist), but sometimes I experience the conflict of these roles", support: 0.06, confidence: 0.45, lift: 2.02.

28. If the female respondent indicated the answer"financial conflict–definitely yes/strongly agree", she also answered:"time conflict—definitely yes/strongly agree","I combine my maternal role with my professional role (woman scientist), but sometimes I experience the conflict of these roles", support: 0.10, confidence: 0.45, lift: 2.01.

**The answer to the research question no. 3.**    The application of the basket analysis algorithm on the data derived from the empirical study made it possible to find real correlations, confirmed by "traditional" methods, between factors characterising the respondents and the types and dimensions of conflict occurring between a scientific career and motherhood as declared by them. Hereby, it can be concluded that a properly configured basket analysis algorithm can be used as a tool supporting the search for correlations not only in a business context, but also in a typically social study.

## Discussion

Summing up the parental and professional situation of the respondents, it should be noted that the vast majority of them declare they experience some kind of conflict between the roles of mother and scientist (81.7%), while most often they indicate that combining the role of mother with the professional role they"sometimes" feel a conflict of these roles (45.8%). The higher the woman is in the scientific hierarchy in terms of degree/title, the better she manages to reconcile her maternal role with her professional role/role as a scientist—and therefore vice versa—the lower the researcher is in the hierarchy, the more often she reports feeling the mother-scientist conflict.

With regard to the dimensions of the mother-scientist role conflict that the respondents currently experience or experienced in the past, the most frequently declared dimension is the time-related one—it concerns or concerned the majority of the respondents (79.3%). In the second place, the respondents indicated the emotional dimension—it concerns or concerned 72.7% of them. The financial dimension of the conflict between research career and motherhood concerns or concerned half of the respondents (49.6%).

The obtained results indicate that the lower a woman is in the scientific hierarchy in terms of rank/title, the more often she declares to feel the time-related dimension of the conflict of roles of mothers-scientist (in the group of respondents there is not a single female Master's degree holder who does not experience this dimension). In the statements of female scientists, the discussed dimensions often appear together, i.e., female scientists often experience more than one dimension of the conflict at once.

The application of traditional academic methodology made it possible to obtain results confirming the existence of conflict and its three types in the opinion of the researched female scientists—mothers, which is consistent with the results of similar, few works in this research area [7]. As for the use of the data mining tool, there have been no similar works; thus it is not possible to compare this study with other ones. To the authors' best knowledge, this has been the first experiment of such kind—possibly other authors, for whom this research will be an inspiration to undertake research and scientific exploration, will refer to the results presented by this paper in the future.

### Threats to validity and study limitations

The study came with a number of factors which might be considered to be limitations to it. The most significant factor were the criteria for the selection of respondents for the study.

They can be considered as limitations of this study, as it only considered the respondents of one gender–i.e. women only. All the respondents belonged to the same professional group and had specific job titles/degrees/academic titles (M.Sc., Ph;). Moreover, all the respondents have been employed at a position which obliges to conduct research and development activity at an academic institution in Poland (excluding teaching positions, which do not oblige to conduct research and development work). All the respondents were employed at the same type of workplace, i.e. academic universities. Lastly, the study only concerned the respondents who were willing to take part in the research. The authors of this work are aware of those possible limitations of the study and will try to compare the results of the respondents belonging to other groups, such as scientist who play the role of a father. The authors would also like to conduct a similar study in other European countries, as it could be beneficial to check and compare if the situation of female scientists is similar there, or not.

In addition to this, as the apriori algorithms tends to be slow and take much computation, other, more efficient data mining algorithms could be used in the future for the analyses of the gathered data.

It must also be noted, that as this study has been one of the numerous ones concentrating specifically on the conflict between the roles of a mother and a scientist to date, and the only one applying basket analysis to analyse this conflict, the literature to support this study's results is still scarce.

## Conclusions

This paper has discussed the selected part of a larger study of mothers-scientists, which concentrated on the conflict of roles and its dimensions.

In the course of the study described in this paper, the answers to all the Research Questions were found:

- The answer to the research question 1.: Do the surveyed women employed at academic institutions in Poland experience conflict between scientific career and motherhood? If so, what kind of conflict?

As it was uncovered, most of the respondents declare they experience the conflict between the roles of a mother and a scientist, to a varying degree. The intensity of the perceived conflict tends to be related to the scientific degree the woman holds–the lower the degree, the more likely she is to experience severe conflict.

- The answer to research question 2.: What are the dimensions of the conflict between academic career and motherhood, according to the surveyed women employed at academic institutions in Poland?

With regard to the dimensions of the mother-researcher role conflict that the respondents currently concern or concerned in the past, the most frequently declared dimension is the time-related one, then subsequently the emotional dimension, and lastly the financial dimension. A substantial number of female scientists declare that they experience more than one dimension of conflict.

- Research question 3.: Is the basket analysis method able to find possible correlations between the factors characterising the respondents and the types and dimensions of the conflict between academic career and motherhood they claim to experience?

The results of the study were then analysed using the basket analysis tool–the apriori algorithm—which objectively confirmed the occurrence of correlations between the factors

characterising the respondents and the types and dimensions of conflict occurring between their scientific career and motherhood. This confirms the usefulness of the tool beyond its typical, traditional use in business—as in this case, in the study of a large social group.

## Author Contributions

**Conceptualization:** Renata Tomaszewska.

**Data curation:** Ewa Krause.

**Formal analysis:** Renata Tomaszewska.

**Funding acquisition:** Ewa Krause, Renata Tomaszewska.

**Investigation:** Ewa Krause.

**Software:** Aleksandra Pawlicka.

**Supervision:** Renata Tomaszewska.

**Writing – original draft:** Aleksandra Pawlicka.

**Writing – review & editing:** Renata Tomaszewska, Aleksandra Pawlicka.

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
