## [Decision Letter · Decision Letter 0]

14 Mar 2022

PONE-D-21-40342Conflicting ’mother-scientist’ roles. An innovative application of basket analysis in social research.PLOS ONE

Dear Dr. Pawlicka,

Thank you for submitting your manuscript to PLOS ONE. After careful consideration, we feel that it has merit but does not fully meet PLOS ONE’s publication criteria as it currently stands. Therefore, we invite you to submit a revised version of the manuscript that addresses the points raised during the review process.

We look forward to receiving your revised manuscript.

Kind regards,

Professor Dr. Mehmet Serkan Kirgiz

Academic Editor

PLOS ONE

Journal Requirements:

3. "Please ensure that you refer to Figure 1 in your text as, if accepted, production will need this reference to link the reader to the figure.

Additional Editor Comments (if provided):

Authors must revise their paper to the comments of reviewer before re-submitting it to PLOS ONE.

Reviewers' comments:

Reviewer's Responses to Questions

**Comments to the Author**

1. Is the manuscript technically sound, and do the data support the conclusions?

Reviewer #1: No

2. Has the statistical analysis been performed appropriately and rigorously? 

Reviewer #1: Yes

3. Have the authors made all data underlying the findings in their manuscript fully available?

Reviewer #1: Yes

4. Is the manuscript presented in an intelligible fashion and written in standard English?

Reviewer #1: No

5. Review Comments to the Author

Reviewer #1: In this paper, the authors set out to discuss the issue of career and motherhood of female scientists, and the paper presents the results of empirical research conducted on the basis of classical academic methodology.

I appreciate the work that has been put into this paper; the heart of this paper is a nice experimental design. However, the paper has major issues listed below:

• It is not clear that the authors follow the introduction guidelines since it does not meet the journal submission guidelines provides requirements about what each part of the manuscript should do (see https://journals.plos.org/plosone/s/submission-guidelines#loc-parts-of-a-submission).

• In particular, the introduction should:

* Provide background that puts the manuscript into context and allows readers outside the field to understand the purpose and significance of the study

* Define the problem addressed and why it is important

* Include a brief review of the key literature

* Note any relevant controversies or disagreements in the field

* Conclude with a brief statement of the overall aim of the work and a comment about whether that aim was achieved.

The seventh publication criterion requires that articles adhere to appropriate reporting guidelines, which include the above guidelines (https://journals.plos.org/plosone/s/criteria-for-publication#loc-7).

• It is not clear that the introduction meets the above requirements. As stated at the end of your conclusion, the theoretical frame, purpose, and research hypothesis will be discussed in later sections of the paper. Although one should expect to read in detail about these aspects of your study in later sections of the paper, it is not clear that the introduction contains sufficient information or rationale about them, nor does it contain a brief statement of the overall aim of the work and a comment about whether that aim was achieved.

• The manuscript does not carefully evaluate the literature on women's rule and its importance, and there are no cites for studies that consider the context of this manuscript.

• The literature should discuss relevant studies about women's social roles to a sufficient depth. For example, how are you women in your cultural context different/the same as how women work in the cultural contexts of previously published studies? Why do we need this study? What hole does it fill?

• The paper suffers from a robust evaluation of the studies and theories that explain the social role and mechanisms behind delays in women's scientific career development.

• The paper does not elaborate the problem well by citing several studies and chaining them to justify the aim of this research.

• A clear theoretical framework could improve the introduction section and provide main insights to the discussion section.

• The discussion does not explain how the paper's findings help address the broad issues raised in the introduction and why this work agrees or disagrees with similar work.

• The discussion suffers from citing other studies to distinguish between opinion and empirical evidence.

• The paper doesn't explain how the limitations of this study.

• I suggest authors add keywords that adequately reflect the paper.

6. PLOS authors have the option to publish the peer review history of their article (what does this mean?). If published, this will include your full peer review and any attached files.

Reviewer #1: No

---

## [Author Response · Author response to Decision Letter 0]

14 Apr 2022

Answers to the Comments concerning:

PONE-D-21-40342

Conflicting ’mother-scientist’ roles. An innovative application of basket analysis in social research.

Dear Reviewers and Editor,

We are very thankful for reading our paper and giving so much valuable feedback. It made us really happy that you found our work interesting. We would like to thank you for your positive approach towards our paper. In this revision, we have implemented all the suggested alterations and we hope that the paper can now be accepted for publication in Plos ONE. 

In particular, we have reacted to the reviews and implemented the following changes:

Editor’s suggestions:

Thank you for this feedback. We’ve adhered the paper to the Word template.

The appropriate information has now been added in the Methods section.

3. "Please ensure that you refer to Figure 1 in your text as, if accepted, production will need this reference to link the reader to the figure.

Thank you for pointing this out! We’ve added the reference to the figures.

Reviewers' suggestions:

Reviewer #1: In this paper, the authors set out to discuss the issue of career and motherhood of female scientists, and the paper presents the results of empirical research conducted on the basis of classical academic methodology.

I appreciate the work that has been put into this paper; the heart of this paper is a nice experimental design. 

Thank you very much for the positive feedback and the heartwarming words!

However, the paper has major issues listed below:

• It is not clear that the authors follow the introduction guidelines since it does not meet the journal submission guidelines provides requirements about what each part of the manuscript should do (see https://journals.plos.org/plosone/s/submission-guidelines#loc-parts-of-a-submission).

Thank you for this feedback!

We’ve re-built the article according to the Plos One criteria.

• In particular, the introduction should:

* Provide background that puts the manuscript into context and allows readers outside the field to understand the purpose and significance of the study

* Define the problem addressed and why it is important

* Include a brief review of the key literature

* Note any relevant controversies or disagreements in the field

* Conclude with a brief statement of the overall aim of the work and a comment about whether that aim was achieved.

The seventh publication criterion requires that articles adhere to appropriate reporting guidelines, which include the above guidelines (https://journals.plos.org/plosone/s/criteria-for-publication#loc-7).

• It is not clear that the introduction meets the above requirements. As stated at the end of your conclusion, the theoretical frame, purpose, and research hypothesis will be discussed in later sections of the paper. Although one should expect to read in detail about these aspects of your study in later sections of the paper, it is not clear that the introduction contains sufficient information or rationale about them, nor does it contain a brief statement of the overall aim of the work and a comment about whether that aim was achieved.

Thank you for these specific requirements! We’ve added the relevant paragraphs in the Introduction section:

Both women and men take on and perform a variety of social roles in society. Women generally function within two main social roles: family and work. The former includes the mother/maternal role and the latter is related to work and career - in the case of the women scientists, to a scientific career. 

Family and professional roles of women are often considered in parallel in order to highlight the difficulties and problems faced by professional women.

(…)

Scientific career is a type of professional career. Even a brief review of the most important literature on this issue indicates the disproportion in reaching its highest stages, compared to men. The reason for this state is mainly motherhood. According to the literature search: 

• For the vast majority of mothers-scientists, motherhood is a factor that significantly modifies (delays) their plans related to the development of a scientific career ;

• Motherhood is the most important factor causing women to withdraw from scientific careers or limiting their pace (1);

• Women scientists' decisions to procreate are often postponed for years, especially after obtaining a doctoral or postdoctoral degree, which often means embarking on late motherhood with the risk of adverse consequences (2);

• Women who are successful in science and have achieved a stable position in academia are more likely than men to be single and childless (3).

As shown later in the paper, although the number of women in Europe pursuing scientific careers is slowly increasing, they are still significantly underrepresented and their potential is not fully recognized and valued. This also applies to Poland.

(…)

The essence of the research problem is reflected in the "Research questions" section:

• Research question 1. Do the surveyed women employed at academic institutions in Poland experience conflict between scientific career and motherhood? If so, what kind of conflict?

• Research question 2. What are the dimensions of the conflict between academic career and motherhood, according to the surveyed women employed at academic institutions in Poland?

• Research question 3. Is the basket analysis method able to find possible correlations between the factors characterising the respondents and the types and dimensions of the conflict between academic career and motherhood they claim to experience?

(…)

It is worth noting that the above-mentioned review of the literature on this issue indicates only a few comprehensive studies on the scientific careers of women and their motherhood as well as studies containing statistical data depicting the scale of the phenomenon of the conflict between the roles of mother and scientist (which we write about in more detail in the State-of-the-Art – Literature Review section). So far, science has dealt only marginally with the impact of motherhood on the careers of women scientists, as well as with the effects of this impact from the point of view of both the researchers and the universities that employ them. Therefore, any attempt to include this issue in the scientific discourse is legitimate and necessary - especially in the form of scientific studies based not only on theoretical analyses, but primarily on the results of empirical research - and this is the case with this article. The presented research results are significant as they allow one to learn about the situation and experiences of female scientists in the context of combining motherhood and scientific career; to show their perspective. Importantly, the situation of women in academia in Poland is comparable to that of women in other European countries, which is discussed further in the article. Therefore, the results presented in this paper constitute a basis for further research analyses conducted by the authors in this problem area and may serve as a source of inspiration for other national and foreign researchers, e.g. in the field of identifying solutions limiting the conflict between the roles of mother and scientist that can be implemented at universities and support women in combining scientific development with fulfilling their parental role. Continuation of the research is needed especially due to the phenomenon of women disappearing/leaking out at successive stages of scientific careers ("leaky pipeline") and thus limiting the waste of their potential and talents. The presence of women in the sector of science and higher education, including in positions which enable making important decisions, is necessary to maintain the effectiveness of the education and science system, as well as to maintain high standards of scientific research. Therefore, it is necessary to diagnose the combination of a scientific career and motherhood not only on a national, but also on a European scale, and to introduce supporting solutions, requiring the needs of female scientists and mothers-researchers to be taken into account by the universities employing them.

The main contribution of this paper, which presents answers to three research questions (the aim of the study was achieved), is that apart from the classical academic methodology, i.e., the following methods: critical analysis of the literature, diagnostic and statistical survey, it has used interdisciplinary methodology supported by data mining techniques. The combination of traditional academic research methodology with the use of data-mining tools created an innovative and original experimental setup. In order to make the study replicable, the specific setup and description of the dataset is given. The presented experiment with the use of the data-mining technique is innovative, as no attempts to apply the so-called basket analysis to the issues of women's careers and motherhood have been reported to date. This study was the first of its kind in Poland. Also, no data on similar studies from other countries in the world have been found.

• The manuscript does not carefully evaluate the literature on women's rule and its importance, and there are no cites for studies that consider the context of this manuscript.

• The literature should discuss relevant studies about women's social roles to a sufficient depth. For example, how are you women in your cultural context different/the same as how women work in the cultural contexts of previously published studies? Why do we need this study? What hole does it fill?

The relevant background has been added:

Although the number of women in Europe pursuing research careers is slowly increasing, they are still significantly underrepresented and their potential is not fully recognized and valued. According to the She Figures 2018 report (29), women represent just over a third (33.4%) of the total researcher population and their proportion in higher education institutions in Europe is only 22%. The latest data from the She Figures 2021 report(30) shows a similar percentage of 32.8% and a share of 23.6% in the leadership of higher education institutions (refers to the EU average). 

Detailed data in relation to selected EU countries are graphically presented in Fig.2 below:

Figure 2. Share of women in the science and research sector in selected EU countries (2021). Source: She Figures 2021, after: A. Pépin, Webinar: "Gender equality in European Research, Innovation and Higher Education. How to enhance scientific excellence through Gender Equality Plans?", 10.03.2022

Data from the She Figures 2021 report also show that although the percentage of women holding the academic title of professor in the EU is gradually increasing, they constitute only slightly more than 1/4 of the total number of its recipients, as presented in Fig. 3.

Figure 3. Percentage of female professors in selected EU countries (2007, 2013, 2016, 2018)

Source: She Figures 2009, 2015, 2018, 2021, after: A. Knapińska, Women in science in Poland. Changing rules of the game, Webinar: "Gender equality in European Research, Innovation and Higher Education. How to enhance scientific excellence through Gender Equality Plans?", 10.03.2022.

Thus, women in comparison to men are less likely to reach the highest stages of the scientific hierarchy in the academic sector (only Romania is balanced in this aspect and Latvia comes close).

In Poland, the situation of women in academia is comparable to that in other countries - their different political past or cultural differences are not relevant. Despite the political changes since the beginning of the 1990s, the transition from a centralized to a free market economy, the rapid massification of higher education, the increasing predominance of women among students, the percentage of women obtaining higher positions in the academic hierarchy is growing very slowly, although steadily. However, in Poland, three times fewer women reach higher levels of scientific careers than men (only 24.7% of people with the academic title of professor are women). The higher the scientific degree/title, the fewer women there are (National Information Processing Institute, National Research Institute 2018). The data has been presented in Table 1.

Table 1. Table Scientific/research and research/research and teaching staff of academic institutions in Poland. Source: compilation by E. Krause based on data provided to the author by OPI PIB (as of 29.01.2018).

Scientific/research and research and teaching staff of academic institutions 

MEN 

WOMEN

Degree/ title total % 

total number % 

total 

of 

100% men: %

of those with the degree/title number % 

total 

of 100% women:

 %

of those with the degree/title

 70320, 

including 100%, 

including 39367, 

including 56, 

including 30953, including: 44, 

including: 

MA 8354 11.9 4095 5.8 10.4 49 4259 6.1 13,8 51

PhD 35023 49.8 17361 24.7 44.1 49.6 17662 25.1 57,1 50.4

PhD hab. 17640 25.1 10910 15.5 27.7 61.8 6730 9.5 21,7 38.2

Professor 9303 13.2 7001 10 17.8 75.3 2302 3.3 7.4 24.7

• The paper suffers from a robust evaluation of the studies and theories that explain the social role and mechanisms behind delays in women's scientific career development.

We’ve added the following background:

It is also worth pointing out the main mechanisms of delays in the development of women's scientific careers, which may provide an interpretative background for the causes of gender imbalance in achieving successive stages of scientific careers. The literature on the subject points to many causes of gender imbalance in science and their different types, e.g. non-institutional and institutional, among which we can distinguish biological and cultural differences or historical background (43)(44). The situation of female researchers in the science and higher education sector can be analysed in the context of mechanisms determining (limiting, hindering) women's professional and scientific careers. They are related to such phenomena as 

- glass ceiling - it is a phenomenon describing invisible obstacles standing in the way of women wishing to advance and thus hindering their professional careers and associated with the occurrence of professional discrimination of vertical/vertical nature. Women are unable to climb the career ladder and reach the highest levels of management

- glass cliff - a phenomenon denoting situations in which a woman's work in a "masculine" (high, decision-making) position (high, decision-making) position is unstable, difficult to maintain and exposed to constant criticism

- glass escalator effect - a discriminatory mechanism in the system of science and higher education, which refers to an invisible force that elevates men to higher levels of professional and scientific careers. It functions on the basis of a stereotypical belief that men (as a group) take decisions and positions that require it more easily and are more suitable for them than for women

- the runaway ladder - this term can be associated with a lack of solidarity among women. Those who are determined and strong pursue their careers through individual strategies. Men, on the other hand, help, support and employ each other. For example, a female scientist may be refused employment or promotion by a female superior or a woman of prominence

- glass walls - another type of horizontal/horizontal occupational discrimination, which means that women do not have access to prestigious, well-paid jobs, industries or positions, and therefore find employment in auxiliary sectors 

- sticky floor - this phenomenon represents situations in which women are "stuck" to their professions and positions - they dominate in professions with low prestige and low income, where there are no opportunities for advancement. In relation to female scientists, it means that more often than their male colleagues they are involved in teaching, which is perceived as less prestigious than conducting research

- velvet ghetto - a term used to describe the fact that women are hindered or precisely prevented from holding managerial and decision-making positions, including positions related to technical sciences, production, marketing. It is also characterized by disrespectful treatment of women holding managerial positions in industries or professions traditionally assigned to men. 

These phenomena, present in academic circles, do not facilitate professional advancement of women in the sector of science and higher education. And as it has already been pointed out in the article - the higher the level of scientific career, the fewer women there are. Few of them also reach the highest positions in the professional hierarchy. Thus, women disappear somewhere "along the way" at different stages of scientific career. It is helpful here to describe further phenomena such as the vanish box and the leaky pipeline. These terms are related to each other. "Magic vanish box" is in fact the term used to describe the "disappearance" of women from the scientific world and their entry (at that time) into alternative career paths. Women's talents "leak out" as they move up the scientific career ladder. In this way, women's talents are wasted, yet qualifications and competencies have no gender. Wasting their potential is particularly noticeable in the sphere of science and higher education. And through the "leaky pipeline" talented female scientists gradually leak out, limiting or completely suspending their scientific and professional development. They "drop out" from higher education institutions and devote themselves to activities other than a scientific career. Family life, especially motherhood, is largely responsible for this "drop out" or "leakage". Most female scientists leave the science and higher education sector between the ages of 30-39. This is because this is the period when they are (usually) heavily involved in caring for and raising children. Previous research has revealed that for the vast majority of scientist mothers, motherhood is a factor that delays their plans for a career in science.

• The paper does not elaborate the problem well by citing several studies and chaining them to justify the aim of this research.

• A clear theoretical framework could improve the introduction section and provide main insights to the discussion section.

• The discussion does not explain how the paper's findings help address the broad issues raised in the introduction and why this work agrees or disagrees with similar work.

• The discussion suffers from citing other studies to distinguish between opinion and empirical evidence.

The relevant comments have been added to the Discussion section.

• The paper doesn't explain how the limitations of this study.

The Threats to Validity section has been added.

• I suggest authors add keywords that adequately reflect the paper.

Thank you for this suggestion! We’ve included the keywords in the revised version: social role, woman, mother, motherhood, scientific career, scientist, conflict, basket analysis, innovation, social research.

Once again, many thanks for the time of the Editor and Reviewers, their careful reading and valuable comments.

We hope that the paper can now be accepted and published, and we hope it will ‘resonate’

within the scientific community.

---

## [Decision Letter · Decision Letter 1]

14 Jun 2022

PONE-D-21-40342R1Conflicting ’mother-scientist’ roles. An innovative application of basket analysis in social research.PLOS ONE

Dear Dr. Pawlicka,

Thank you for submitting your manuscript to PLOS ONE. After careful consideration, we feel that it has merit but does not fully meet PLOS ONE’s publication criteria as it currently stands. Therefore, we invite you to submit a revised version of the manuscript that addresses the points raised during the review process.

Your manuscript has been scientifically judged for a second time, finding out that our reviewer still raises (among others) several concerns over the technical background and data interpretation of the manuscript. Nwevertheless, and since I personally believe this paper has potential, IO would like to give you the chance to re-revise the paper, in order to reconsider the editorial decision.

We look forward to receiving your revised manuscript.

Kind regards,

Sergio A. Useche, Ph.D.

Academic Editor

PLOS ONE

Journal Requirements:

Additional Editor Comments (if provided):

Reviewers' comments:

Reviewer's Responses to Questions

**Comments to the Author**

1. If the authors have adequately addressed your comments raised in a previous round of review and you feel that this manuscript is now acceptable for publication, you may indicate that here to bypass the “Comments to the Author” section, enter your conflict of interest statement in the “Confidential to Editor” section, and submit your "Accept" recommendation.

Reviewer #1: (No Response)

2. Is the manuscript technically sound, and do the data support the conclusions?

Reviewer #1: Partly

3. Has the statistical analysis been performed appropriately and rigorously? 

Reviewer #1: Yes

4. Have the authors made all data underlying the findings in their manuscript fully available?

Reviewer #1: No

5. Is the manuscript presented in an intelligible fashion and written in standard English?

Reviewer #1: No

6. Review Comments to the Author

Reviewer #1: Thank you for submitting the revised copy of the manuscript; I do find the manuscript very interesting and the topic worth being published; However, the manuscript suffers from significant issues and I believe it can not be accepted in this format:

Still, the theoretical framework is unclear; I suggest having a separate section for this and supporting it with a diagram presenting the different variables.

I have serious concerns about the study validity and reliability assessment and analysis; please clarify them in a separate section how you did test the questionnaire validity and reliability in the method section.

Still, there are no cites for studies that consider the context of this manuscript, especially the theoretical framework.

You did have the part on the mechanisms behind delays in women's scientific career development; please have it as a separate section and expand on it supported with references.

The discussion suffers from citing other studies to distinguish between opinion and empirical evidence; the authors did not elaborate enough on this point.

The discussion did not discuss the results raised by each research question presented in the study.

I highly recommend the authors get assistance from an English professional writer since there are a lot of type and grammar mistakes and unclear sentences.

The structure of the paper needs a lot of work so the manuscript can sound publishable.

I placed comments on the manuscript; Please refer to it as well and response in a table.

If the authors would like to apply the changes: please place the comments (above and in the manuscript) and the response in a table.

Having the comments and the response been made with a page number it will be much easier to follow.

7. PLOS authors have the option to publish the peer review history of their article (what does this mean?). If published, this will include your full peer review and any attached files.

Reviewer #1: **Yes: **Firas Almasri

---

## [Author Response · Author response to Decision Letter 1]

27 Jul 2022

* The full, comprehensive response with color coded changes and answers has been attached in he file inventory. *

---

## [Decision Letter · Decision Letter 2]

20 Sep 2022

PONE-D-21-40342R2Conflicting ’mother-scientist’ roles. An innovative application of basket analysis in social research.PLOS ONE

Dear Dr. Pawlicka,

Thank you for submitting your manuscript to PLOS ONE. After careful consideration, we feel that it has merit but does not fully meet PLOS ONE’s publication criteria as it currently stands. Therefore, we invite you to submit a revised version of the manuscript that addresses the points raised during the review process. Your paper has been re-reviewed. Overall, our referee suggests the consideration of your manuscript for acceptance, but only after a new phase of revisions, in which some minor (but still relevant) points must be addressed by you to the best of the possibilities. Please find the full set of comments below. 

We look forward to receiving your revised manuscript.

Kind regards,

Sergio A. Useche, Ph.D.

Academic Editor

PLOS ONE

Journal Requirements:

Reviewers' comments:

Reviewer's Responses to Questions

**Comments to the Author**

1. If the authors have adequately addressed your comments raised in a previous round of review and you feel that this manuscript is now acceptable for publication, you may indicate that here to bypass the “Comments to the Author” section, enter your conflict of interest statement in the “Confidential to Editor” section, and submit your "Accept" recommendation.

Reviewer #2: All comments have been addressed

2. Is the manuscript technically sound, and do the data support the conclusions?

Reviewer #2: Yes

3. Has the statistical analysis been performed appropriately and rigorously? 

Reviewer #2: Yes

4. Have the authors made all data underlying the findings in their manuscript fully available?

Reviewer #2: Yes

5. Is the manuscript presented in an intelligible fashion and written in standard English?

Reviewer #2: Yes

6. Review Comments to the Author

Reviewer #2: Dear Authors,

First of all, thanks for contributing with this interesting research about “Conflicts mother’s scientist’s role “After a careful assessment of the paper, I believe the reviewed manuscript addresses a pertinent research problem (i.e. female scientists and motherhood) for being considered as publishable in PLOS ONE. Of course, some points need to be addressed beforehand.

In general, the structure of the paper is adequate; however, the presentation can be clearer and more concise. In this regard, some concerns, queries and suggestions raised during my review must be addressed, in order to optimize the manuscript contents and its suitability for the journal:

• ABSTRACT: The relation between these concepts is not clear:

Please introduce the results in the abstract

• METHODS: For a better understanding of the study, please include a section where each of the methods used can be clearly described.

It would be advisable to describe the process for answer the questions better for the sample

• CONCLUSION: Adequate, and within the scope of the data. However, study limitations could be improved.

I congratulate you for your perseverance and effort made to carry out this work

7. PLOS authors have the option to publish the peer review history of their article (what does this mean?). If published, this will include your full peer review and any attached files.

Reviewer #2: No

---

## [Author Response · Author response to Decision Letter 2]

21 Sep 2022

The file with the full answer has been included in this revision.

---

## [Editor Report · Decision Letter 3]

2 Oct 2022

Conflicting ’mother-scientist’ roles. An innovative application of basket analysis in social research.

PONE-D-21-40342R3

Dear Dr. Pawlicka,

We’re pleased to inform you that your manuscript has been judged scientifically suitable for publication and will be formally accepted for publication once it meets all outstanding technical requirements.

Kind regards,

Sergio A. Useche, Ph.D.

Academic Editor

PLOS ONE
---

## [Editor Report · Acceptance letter]

10 Oct 2022

PONE-D-21-40342R3 

Conflicting ’mother-scientist’ roles. An innovative application of basket analysis in social research. 

Dear Dr. Pawlicka:

I'm pleased to inform you that your manuscript has been deemed suitable for publication in PLOS ONE. Congratulations! Your manuscript is now with our production department. 

Kind regards, 

on behalf of

Dr. Sergio A. Useche 

Academic Editor

PLOS ONE